# Risk Factors Associated with Dengue Virus Infection in Guangdong Province: A Community-Based Case-Control Study

**DOI:** 10.3390/ijerph16040617

**Published:** 2019-02-20

**Authors:** Jundi Liu, Xiaolu Tian, Yu Deng, Zhicheng Du, Tianzhu Liang, Yuantao Hao, Dingmei Zhang

**Affiliations:** School of Public Health, Sun Yat-sen University, Guangzhou 510080, China; liujd5@mail2.sysu.edu.cn (J.L.); tianxlu@mail2.sysu.edu.cn (X.T.); dengyu9@mail.sysu.edu.cn (Y.D.); duzhch3@mail2.sysu.edu.cn (Z.D.); tianzhu@bu.edu (T.L.); haoyt@mail.sysu.edu.cn (Y.H.)

**Keywords:** dengue fever, Aedes albopictus, living environment, logistic regression analysis

## Abstract

Dengue fever (DF) is a mosquito-borne infectious disease that is now an epidemic in China, Guangdong Province, in particular and presents high incidence rates of DF. Effective preventive measures are critical for controlling DF in China given the absence of a licensed vaccination program in the country. This study aimed to explore the individual risk factors for the dengue virus infection in Guangdong Province and to provide a scientific basis for the future prevention and control of DF. A case-control study including 237 cases and 237 controls was performed. Cases were defined for samples who were IgG-antibody positive or IgM-antibody positive, and willing to participate in the questionnaire survey. Additionally, the controls were selected through frequency matching by age, gender and community information from individuals who tested negative for IgG and IgM and volunteered to become part of the samples. Data were collected from epidemiological questionnaires. Univariate analysis was performed for the preliminary screening of 28 variables that were potentially related to dengue virus infection, and multivariate analysis was performed through unconditioned logistic regression analysis to analyze statistically significant variables. Multivariate analysis revealed two independent risk factors: Participation in outdoor sports (odds ratio (OR) = 1.80, 95% confidence interval (CI) 1.17 to 2.78), and poor indoor daylight quality (OR = 2.27, 95% CI 1.03 to 5.03). Two protective factors were identified through multivariate analysis: 2 occupants per room (OR = 0.43, 95% CI 0.28 to 0.65) or ≥3 occupants per room (OR = 0.45, 95% CI 0.23 to 0.89) and air-conditioner use (OR = 0.46, 95% CI 0.22 to 0.97). The results of this study were conducive for investigating the risk factors for dengue virus infection in Guangdong Province. Effective and efficient strategies for improving environmental protection and anti-mosquito measures must be provided. In addition, additional systematic studies are needed to explore other potential risk factors for DF.

## 1. Introduction

Dengue fever (DF) is an acute viral disease caused by four distinct serotype dengue viruses, transmitted between humans by the mosquito *Aedes aegypti*. Statistics provided by the World Health Organization (WHO) has shown that only nine countries experienced severe dengue epidemics before 1970. Recent research, however, has revealed that the number of countries with severe dengue epidemics now exceeds 100 and that the actual number of DF cases has reached approximately 390 million, among which 500,000 patients required hospital admission because of severe infection [1]. In Asian and American countries wherein dengue is endemic, the effect of dengue is approximately 1300 disability-adjusted life years per million population; this effect is highly similar to the disease burden of related childhood and tropical diseases, including tuberculosis [2]. 

In Asia, the coverage of epidemic dengue hemorrhagic fever (DHF) has expanded geographically westward from southeast Asian countries to India, Sri Lanka, the Maldives, and Pakistan and then eastward to China [2]. In China, the first reported DF outbreak due to dengue virus type 4 occurred in Foshan City, Guangdong Province, in 1978; DF then began to spread to southern Chinese provinces from Foshan City [3]. Since then, Guangdong Province has exhibited the highest incidence of DF in mainland China, and more than 65% of all DF cases in the country were reported in this province [4]. In 2014, the number of DF cases increased dramatically to 38,753 in Guangdong province and accounted for 93.83% of DF cases in mainland China [5]. *Aedes albopictus*, an aggressive mosquito species that is also one of the main vectors of DF, is widely distributed with high density in Guangdong Province [6,7]. Therefore, controlling the outbreak of DF in Guangdong Province, which can act as a bridge for DF transmission to other provinces in mainland China, is urgent. Unfortunately, effective drugs and a licensed vaccination program for the treatment or prevention of DF are unavailable in China. 

Understanding the risk factors for dengue virus infection is necessary to control this disease effectively. However, most of the current case–control studies on risk factors for DF focused on severe dengue infections, such as dengue shock syndrome and DHF, and variables related to clinical and laboratory indexes [8,9,10,11]. Environmental factors, such as heavy rainfall and global warming, and factors based on the awareness and knowledge of dengue prevention measures are also responsible for drastic reductions in dengue transmission [12,13]. Several macroscopic descriptive studies have been performed to explore the risk factors for dengue virus infection and to provide a basis for formulating control strategies in Guangdong Province. These studies have obtained considerable information on the group level and climate factors but limited information on personal protective measures [14,15]. 

In this case-control study, we evaluated potential risk factors, including personal life activities, environmental sanitation, housing situation, living conditions, mosquito protection status, and residential surroundings to identify additional risk factors for DF on the individual level and to recommend specific approaches for preventing DF. 

## 2. Materials and Methods

### 2.1. Community Selection and Study Design 

Guangzhou City and Zhongshan City is located in the Pearl River Delta Region of Guangdong, which is the main area where DF is highly epidemic [16,17]. Guangzhou City is the capital of Guangdong Province, and the first reported case of autochthonous DF occurred in Zhongshan City [18]. Thus, the prevalence of DF in Guangzhou City and Zhongshan City is a good representation of the prevalence of DF in Guangdong Province.

This case-control study was performed on the basis of the project of Research on the Prevention and Control of Human Immunodeficiency Virus and Hepatitis B Virus in Guangdong Province. This project has constructed a database containing 200,000 samples. The demographic information contained by the database could be seen in our related publication [19]. Approximately 30–35 persons per month were sampled from every age group (<19 years, 19–40 years, 41–65 years, and >65 years) over a 2-year period from September 2013 to August 2015. 

### 2.2. Ethical Statement

This work obtained ethics approval from the Institutional Review Board of the School of Public Health at Sun Yat-sen University (L2017030) in line with the guidelines for the protection of human subjects. All research participants or their guardians provided signed written informed consent after being informed of the research subject matter and were assured that their personal information would be kept private. Each participant had the right to withdraw from this study at any time. 

### 2.3. Enzyme Immunoassay Test

Enzyme-linked immunosorbent assay (ELISA) was used to detect dengue IgG and IgM antibodies. The IgG antibody was measured through indirect ELISA (LOT: 01P20A006, Inverness Medical/Panbio, Windsor, Australia). The IgM antibody was tested via capture ELISA (LOT:01P30A002, Inverness Medical/Panbio, Windsor, Australia). Undefined results were confirmed through the colloidal gold method (LOT: DEN141001, Inverness Medical/Cortez, Calabasas, CA, USA). The details of the assays could be found in a previous publication [19].

### 2.4. Case Definition and Control Selection 

Among the 3136 serum samples, 305 and 103 were identified as IgG-antibody positive and IgM-antibody positive, respectively, and 34 were positive for both antibodies. Thus, 374 individuals with antibody-positive samples were defined as dengue-infected, and those willing to participate in the questionnaire survey were selected as the members of the case group. 

In this study, 256 patients with dengue infections opted to fill in the questionnaires, and 19 questionnaires missing most of the important information were eliminated. Eventually, 237 cases were included in the case group. 

The controls were selected through frequency matching from individuals who tested negative for IgG and IgM. Specifically, candidate controls were stratified in accordance with the age, gender, and community information of the case group and selected through convenience sampling, wherein participants volunteered to become part of the samples, from each layer. Additionally, age matching was carried out according to ≤15 years, 16–30 years, 31–50 years, 51–65 years and ≥66 years. Finally, 237 controls were selected.

### 2.5. Data Collection and Analysis

The phone questionnaire investigation was conducted by trained investigators. Subjects who interrupted the telephone investigation and whose questionnaire information contained logic errors were interviewed face-to-face to verify the integrity and validity of their information. Just as shown in Appendix A, the main contents of the questionnaire included general demographic characteristics (age, gender, blood type, and average household income). It also included personal life activities, such as activities in the park, outdoor sports (such as hiking, mountain climbing, and camping), and outbound tourism experience. Moreover, the questionnaire presented questions related to environmental sanitation (domestic sewage and garbage management and participation in community hygiene management interventions); housing situation, such as the age and area of domiciles and living floor; and living conditions (average numbers of occupants per room, use of air conditioning, quality of indoor daylight, and presence of animals or aquatic plants on property). It also had questions on mosquito protection status (use of mosquito nets and pesticide) and residential surroundings (presence of junk yards, ponds, or construction sites within 200 m).

“Contact with patients with DF” was defined as previous living or working experience with patients with DF in the past years in his/her life. The definition of “outdoor activities in parks” was established as outdoor activity for at least two times per week and for more than half an hour each time. “Participation in outdoor sports” was defined as participation in activities at least twice a year for more than half an hour each time. The definition of “good indoor daylight quality” was established on the basis of the minimum requirement for sunshine in general residences of not less than 2 h on extremely cold days. The definition of “good ventilation” was given to an open-ventilation area of not less than 5% of the floor area of each domicile. The use of mosquito repellent referred to the use of mosquito coils and insecticide vaporizers. “Occasionally” was defined as less than once a week, and “often” was defined as at least once a week.

Epidata 3.1 software (Epidata Association, Odense, Denmark) was used to establish a database of individual risk factors for dengue infection among residents in Guangdong Province. All data were analyzed by SPSS statistics 23.0 software (SPSS Inc., Chicago, IL, USA). The χ^2^ test was used to test for differences in demographic characteristics between cases and controls. A univariate unconditioned logistic regression analysis was applied for the preliminary screening of variables identified by using the questionnaire but not for the variables of general demographic characteristics. An unconditioned logistic regression analysis method for multivariate analysis was employed to analyze the relative importance of statistically significant variables in univariate analysis. Additionally, considering the rule of frequency matching design, the age, gender and community information variables were also introduced. Then, stepwise regression was used to establish a regression equation. *p* < 0.05 was set as the significance level of the χ^2^ test and multivariate analysis. However, to avoid missing important factors, *p* < 0.1 was set as the significance level in univariate analysis. In addition to odds ratios (ORs), 95% confidence intervals (CIs) were used to express associations. 

## 3. Results

### 3.1. General Demographic Characteristics of the Samples 

A total of 3136 serum samples collected from the residents of Yuexiu District in Guangzhou City (*n* = 699), Liwan District in Guangzhou City (*n* = 1386), Torch Development Area in Zhongshan City (*n* = 180), and Xiaolan Town in Zhongshan City (*n* = 871) were selected via stratified cluster sampling rooted in the database for serological testing. The study population had a male: female ratio of 1:1.92. Age statistics showed that the age group of ≥66 years old accounted for the largest proportion of the study population (25.86%), followed by 51–65 years old.

Finally, 474 subjects, including 237 cases and 237 controls, were recruited successfully (Figure 1). No statistical difference in gender (*p* = 0.950) and age (*p* = 0.127) existed between persons who were willing to receive the questionnaire survey and those who were unwilling to receive the questionnaire survey. The gender ratio was 1:1.66 (male: female) in both the case group and the control group. The demographic characteristics of the two groups were comparable (Table 1). 

### 3.2. Univariate Analysis

A total of 28 potential risk factors were analyzed. These factors were further divided into six dimensions: Personal life activities, environmental sanitation, housing situation, living conditions, mosquito protection status, and residential surroundings. As illustrated in Table 2, people who participated in outdoor activities in parks had a significantly higher probability of DF infections than those who did not participate in outdoor activities in parks (*p* = 0.049). People who participated in outdoor sports were more likely to be infected with DF than those who did not participate in outdoor sports (*p* = 0.009). At the same time, there were statistical differences in terms of housing type (*p* = 0.040), housing location (*p* = 0.061), living floor (*p* = 0.096), the average numbers of persons per room (*p* < 0.001), air-conditioner use (*p* = 0.026) and indoor daylight quality (*p* = 0.032) between the case group and the control group. 

The existence of garbage collection sites (*p* = 0.681), junk yards (*p* = 0.570), ponds (*p* = 0.426), and construction sites (*p* = 0.639) within 200 m of residences did not show statistically significant differences between the control and case groups.

### 3.3. Multivariate Analysis

Multivariate logistic regression analysis was performed on the basis of the results of univariate analysis and the rule of frequency matching design. In the unconditioned logistic regression model, participation in outdoor sports (OR = 1.80, 95% CI = 1.17 to 2.78) and poor indoor daylight quality (OR = 2.27, 95% CI = 1.03 to 5.03) were significantly associated with an increased risk of dengue virus infection. On the other hand, 2 occupants per room (OR = 0.43, 95% CI = 0.28 to 0.65), ≥3 occupants per room (OR = 0.45, 95% CI = 0.23 to 0.89), and air-conditioner use (OR = 0.46, 95% CI = 0.22 to 0.97) were significantly associated with protection against dengue virus infection (Table 3). 

## 4. Discussion

We found that participation in outdoor sports activities and poor indoor daylight quality significantly increased the probability of contracting DF by 1.80- and 2.27-fold, respectively, in Guangdong Province. Our results also suggested that 2 occupants per room, ≥3 occupants per room, and air-conditioner use might decrease the probability of dengue virus infection by 0.43-, 0.45-, and 0.46-fold, respectively.

Our study revealed that people who participated in outdoor sports were at a significantly higher risk of contracting DF than those who did not participate in outdoor sports. This result may be attributed to the preference of residents who participate in outdoor sports to hike and camp in forest margins, tree copses, and natural reserves, which are the original habitats of *A. albopictus* [20]. Therefore, given that participation in outdoor sports increases the risk of exposure to mosquito bites, anti-mosquito measures, such as treating outdoor areas and materials with insecticides, must be adopted.

We also found that human population density was closely associated with dengue transmission. In general, high population density is a risk factor for dengue transmission [21,22]. We found, however, that crowded households comprising ≥2 occupants were at low risk of dengue infection. On the contrary, Velascosalas et al. found that crowded households with more than 1.5 occupants in one room were at risk of dengue infection [23]. We conducted our study in communities wherein individuals lived together in family groups and wherein parents and their young children tended to share one room. Wang et al. reported that 62.48% of Chinese children aged 0–5 years old shared beds with their parents [24]. Our result may be attributed to the following: Parents who share rooms with their children pay additional attention to the use of anti-mosquito measures and the maintenance of good environmental sanitation to protect their children from mosquito bites. Our previous study also showed that married participants had a lower rate of infection than widowed and divorced participants [19]. These results suggest that married groups who reside in one room with ≥2 occupants are at a reduced risk of DF infection. Moreover, high numbers of occupants in one room are associated with the decreased probability of mosquito bites when the number of mosquitoes was fixed.

Shen et al. [25] and Wu et al. [26] reported that yearly average temperatures of more than 18 °C would increase the risk of dengue virus infection. Meanwhile, our study indicated that air-conditioner use was a protective factor against dengue infection by reducing the risk of dengue transmission through cooling the indoor environment. In addition, doors and windows are commonly shut during air-conditioner use; this practice could also reduce the chance that mosquitoes could enter the rooms [27].

We found that the poor indoor daylight quality increased the likelihood of infection with DF by 2.27-fold because adult *A. albopictus* prefer to inhabit poorly lit areas over well-lit areas [28,29]. As a result, environments receiving insufficient daylight encourage the density of mosquitoes to increase because they are suitable for the survival of mosquitoes.

Vanwambeke et al. [30] and Kenneson et al. [31] reported that the use of mosquito nets reduced the risk of dengue virus infection. However, similar to Tsuzuki et al. [32] and Loroñopino et al. [33], we failed to find a relationship between the use of mosquito nets and the likelihood of dengue virus infection. The lack of a relationship between this variable and dengue virus infection may be attributed to the following: Mosquito nets are usually used at night. However, *A. albopictus* is active during the day, especially in the early morning and late afternoon [34]. Other studies found that mosquito nets play a protective role in preventing dengue virus infection in rural settings [29]. However, we recruited our study population from urban areas. The good living environment in urban areas, for example, the popularity of air-conditioning and mosquito killing facilities, reduced the demand for mosquito nets. Nevertheless, on the basis of our local experience and some studies’ results [30,35], we recommend using mosquito nets not only at night but also during the day [31].

Andersson et al. [36] and Roberto et al. [11] reported that the government’s ability and capacity to control the dengue vector has crucial effects on dengue transmission. Community neighborhood committees and property management departments in Guangdong Province have organized numerous health remediation activities under the supervision of the relevant health agent or the Centers for Disease Control and Prevention [37] given the high incidence of DF in recent years. These activities have considerably improved the residential living environment and reduced mosquito breeding and may account for the lack of the statistical significance of the variables of domestic sewage disposal, garbage management, and residential surroundings in this study. However, we found that 67.51% of the cases and 60.39% of the controls did not participate in the community hygiene management intervention activities organized by neighborhood committees or property management organizations. The public health consciousness of the case and control groups must be strengthened because the government will be unable to establish a sound prevention system against DF despite having a good macro-control system in place if it lacks the support of the masses.

## 5. Limitations

This study has several flaws that should be overcome in future studies. First, cases and controls were identified in accordance with the results of antibody detection. However, IgG-positive samples may have been derived from patients who were infected with DF several years ago. Thus, the results of the completed questionnaires of these patients may be incompatible with their situations during their initial infection. Second, recall biases stemming from the inaccuracy of memory could also reduce the validity of the questionnaire. Third, newly infected individuals may have been misdiagnosed as controls because IgG and IgM antibody titers are present at undetectable levels during the initial stages of infection with dengue [38]. Thus, misclassification bias may have been introduced. Fourth, volunteer bias might have been caused by participants who volunteered to become part of the samples. Additionally, the information provided by the volunteered participants might have been different from the general population. Fifth, some definitions like the definitions of “good indoor daylight quality” and “good ventilation” were given in the questionnaire, but respondents might have had a subjective understanding of these definitions, which could have led to inaccuracy of the relevant information. Finally, the model’s general applicability was difficult to evaluate at present given that the data used in this study were obtained on the basis of serum samples collected from residents in Guangdong Province over the period of 2013 to 2015 without other extrapolated studies. Given that other studies have shown that the community wherein residents live remains an independent factor associated with DF infection [23], the results of this study could be used as the reference for the development of personalized protective measures against DF infection in tropical and subtropical countries with high densities of *Aedes* mosquitoes.

## 6. Conclusions

Our study focused on residents in communities with mild or asymptomatic dengue virus infection rather than on patients with severe clinical symptoms to explore the risk factors for dengue virus infection in Guangdong Province because the former sample is highly representative. We established the relationship between dengue infection and individual risk factors. This information is beneficial for avoiding infection by the dengue virus. We also provided evidence and a basis for the development of measures for DF prevention and control. Additional variables must be introduced into the logistic regression model, and further research should be conducted to provide a theoretical basis for formulating prevention and control measures for DF. 

## Figures and Tables

**Figure 1 ijerph-16-00617-f001:**
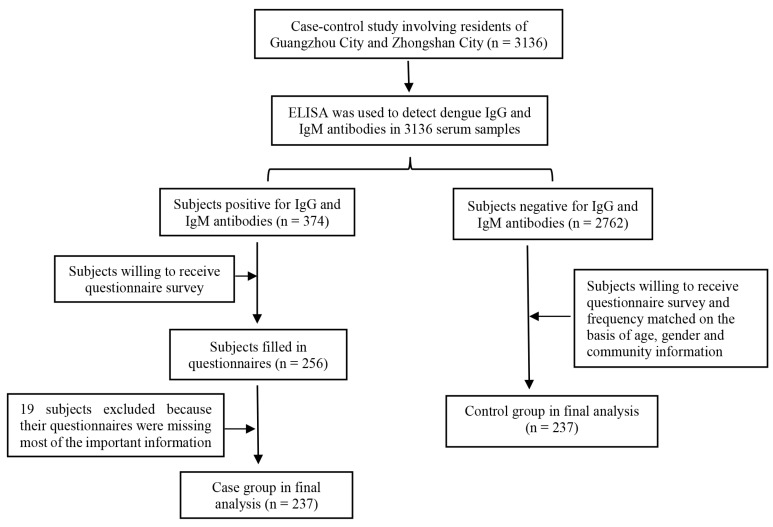
Consort diagram of cases and controls used in the study.

**Table 1 ijerph-16-00617-t001:** Demographic characteristics of cases and controls.

Demographic Characteristics ^a^	Cases (*n*/%) (*n* = 237)	Controls (*n*/%) (*n* = 237)	*p*-Value
Age (years) ≤15 16–30 31–50 51–65 ≥66	15 (6.33) 13 (5.49) 33 (13.92) 43 (18.14) 133 (56.12)	15 (6.33) 13 (5.49) 33 (13.92) 43 (18.14) 133 (56.12)	-
Gender			-
Male	89 (37.55)	89 (37.55)	
Female	148 (62.45)	148 (62.45)	
Residential status			0.149
Permanent residents	225 (94.94)	231 (97.47)	
Floating population ^b^	12 (5.06)	6 (2.53)	
Number of residents per household			0.394
1	27 (11.39)	17 (7.17)	
2–3	126 (53.16)	125 (52.75)	
4–5	75 (31.65)	84 (35.44)	
≥6	9 (3.80)	11 (4.64)	
Monthly per capita family income (¥) ^c^			0.317
<2000	50 (21.10)	36 (15.19)	
2000–4999	145 (61.18)	162 (68.35)	
5000–7999	35 (14.77)	31 (13.08)	
≥8000	7 (2.95)	8 (3.38)	
Blood type			0.196
A	18 (7.59)	14 (5.91)	
B	15 (6.33)	19 (8.02)	
O	29 (12.24)	46 (19.40)	
AB	9 (3.80)	6 (2.53)	
Unknown	166 (70.04)	152 (64.14)	

^a^ Except where otherwise indicated, values are the number (percentage) of patients with the characteristic. ^b^ Floating population refers to migrants who live locally for 6 months or less. ^c^ The minimum wage in Guangzhou City is 2100 yuan per month and the minimum wage in Zhongshan City is 1720 yuan per month.

**Table 2 ijerph-16-00617-t002:** Univariate analysis of risk factors for dengue virus infection.

Variables	Cases (*n*/%) (*n* = 237)	Controls (*n*/%) (*n* = 237)	OR (95% CI)	*p*-Value
Contact with patients with dengue fever				0.655
Yes	3 (1.27)	2 (0.84)	1.51 (0.25–9.10)	
No	234 (98.73)	235 (99.16)	Reference	
Outbound tourism experience				0.867
Yes	19 (8.02)	20 (8.44)	Reference	
No	218 (91.98)	217 (91.56)	1.06 (0.55–2.04)	
Outdoor activities in parks				0.049 *
Yes	200 (84.39)	183 (77.22)	1.60 (1.00–2.54)	
No	37 (15.61)	54 (22.78)	Reference	
Participation in outdoor sports				0.009 *
Yes	74 (31.22)	49 (20.68)	1.74 (1.15–2.64)	
No	163 (68.78)	188 (79.32)	Reference	
Domestic sewage disposal frequency				0.655
Daily	125 (52.74)	118 (49.79)	Reference	
2 days	25 (10.55)	20 (8.44)	1.18 (0.62–2.24)	
≥3 days	35 (14.77)	42 (17.72)	0.79 (0.47–1.32)	
No domestic sewage	52 (21.94)	57 (24.05)	0.86 (0.55–1.35)	
Garbage disposal frequency				0.311
Daily	223 (94.09)	228 (96.20)	Reference	
2 days	11 (4.64)	5 (2.11)	2.25 (0.77–6.58)	
≥3 days	3 (1.69)	4 (1.69)	0.77 (0.17–3.47)	
Participation in community hygiene management interventions				0.104
Yes	77 (32.49)	94 (39.66)	Reference	
No	160 (67.51)	143 (60.34)	1.37 (0.94–1.99)	
Location				0.061 *
Rural	15 (6.33)	11 (4.64)	0.34 (0.08–1.51)	
City	210 (88.61)	223 (94.09)	0.24 (0.07–0.85)	
Urban–rural integration	12 (5.06)	3 (1.27)	Reference	
Housing building structure				0.871
Brick–wood structure	2 (0.84)	3 (1.27)	0.67 (0.11–4.05)	
Brick–wood and concrete structure	31 (13.08)	29 (12.24)	1.07 (0.63–1.85)	
Concrete structure	204 (86.08)	205 (86.49)	Reference	
Housing type				0.040 *
Single-family apartment	60 (25.32)	39 (16.46)	Reference	
Commercial residential community	173 (73.00)	196 (82.70)	0.57 (0.37–0.90)	
Villa	4 (1.68)	2 (0.84)	1.30 (0.23–7.44)	
Housing age (year)				0.919
<10	23 (9.71)	24 (10.13)	Reference	
10–20	115 (48.52)	116 (48.95)	1.03 (0.54–1.94)	
20–40	88 (37.13)	89 (37.55)	1.03 (0.54–1.96)	
>40	11 (4.64)	8 (3.37)	1.43 (0.49–4.21)	
Number of floors per residential structure				0.096 *
1–3	110 (46.41)	87 (36.71)	Reference	
4–9	103 (43.46)	124 (52.32)	0.66 (0.45–0.96)	
≥10	24 (10.13)	26 (10.97)	0.73 (0.39–1.36)	
Average numbers of persons per room				<0.001 *
1	91 (38.40)	51 (21.52)	Reference	
2	123 (51.90)	160 (67.51)	0.43 (0.28–0.65)	
≥3	23 (9.70)	26 (10.97)	0.50 (0.26–0.96)	
Housing area (m^2^)				0.235
<50	63 (26.58)	52 (21.9)	Reference	
51–100	150 (63.29)	169 (71.31)	0.73 (0.18–1.12)	
101–150	19 (8.02)	11 (4.64)	1.43 (0.62–3.26)	
>150	5 (2.11)	5 (2.11)	0.83 (0.23–3.01)	
Air-conditioner use				0.013 *
Never	26 (10.97)	11 (4.64)	Reference	
Yes	211(89.03)	226 (95.36)	0.40 (0.19–0.82)	
Indoor daylight quality				0.032 *
Good	215 (90.72)	227 (95.78)	Reference	
Poor	22 (9.28)	10 (4.22)	2.32 (1.08–5.02)	
Ventilation				0.324
Good	221 (93.25)	226 (95.36)	Reference	
Bad	16 (6.75)	11 (4.64)	1.49(0.68–3.28)	
Keeping of pets				0.800
Yes	38 (16.03)	36 (15.19)	Reference	
No	199 (83.97)	201 (84.81)	0.94 (0.57–1.54)	
Raising of poultry				0.589
Yes	6 (2.53)	8 (3.38)	0.74 (0.25–2.18)	
No	231 (97.47)	229 (96.62)	Reference	
Breeding of aquatic plants				0.578
Yes	54 (20.68)	49 (20.68)	Reference	
No	183 (77.22)	188 (79.32)	0.88 (0.57–1.37)	
Use of mosquito nets				0.361
Yes	185 (78.06)	193 (81.43)	Reference	
No	52 (21.94)	44 (18.57)	1.23 (0.79–1.93)	
Use of mosquito repellent				0.212
Never	122 (51.48)	116 (48.95)	Reference	
Occasionally	88 (37.13)	103 (43.46)	0.81 (0.56–1.19)	
Often	27 (11.39)	18 (7.59)	1.43 (0.75–2.73)	
Use of electric mosquito-killing devices				0.150
Never	152 (64.14)	143 (60.34)	Reference	
Occasionally	57 (24.05)	74 (31.22)	0.73 (0.48–1.10)	
Often	28 (11.81)	20 (8.44)	1.32 (0.71–2.44)	
Use of camphor				0.649
Never	177 (74.68)	168 (70.89)	Reference	
Occasionally	42 (17.72)	48 (20.25)	0.83 (0.52–1.32)	
Often	18 (7.60)	21 (8.86)	0.81 (0.42–1.58)	
Existence of garbage collection sites within 200 m around housing				0.681
Yes	32 (13.50)	29 (12.24)	1.12 (0.65–1.92)	
No	205 (86.50)	208 (87.76)	Reference	
Existence of junk yards within 200 m around housing				0.570
Yes	1 (0.42)	2 (0.84)	0.50 (0.05–5.53)	
No	236 (99.58)	235 (99.16)	Reference	
Existence of ponds within 200 m around housing				0.426
Yes	45 (18.99)	52 (21.94)	0.83 (0.53–1.30)	
No	192 (81.01)	185 (78.06)	Reference	
Existence of construction sites within 200 m around housing				0.639
Yes	24 (10.13)	21 (8.86)	1.16 (0.63–2.15)	
No	213 (89.87)	216 (91.14)	Reference	

* Significance difference: *p* < 0.1.

**Table 3 ijerph-16-00617-t003:** Multivariate analysis of risk factors for dengue virus infection.

Risk factors	Odds Ratio	95% CI	*p*-Value
Participation in outdoor sports			0.007 *
Yes	1.80	(1.17–2.78)	
No	Reference		
Average numbers of occupants per room			<0.001 *
1	Reference		
2	0.43	(0.28–0.65)	<0.001 *
≥3	0.45	(0.23–0.89)	0.021 *
Air-conditioner use			0.040 *
Never	Reference		
Yes	0.46	(0.22–0.97)	
Indoor daylight quality			0.043 *
Good	Reference		
Poor	2.27	(1.03–5.03)	

* Significant difference when *p* < 0.05.

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
