# Peer review of "Risk Factors Associated with Dengue Virus Infection in Guangdong Province: A Community-Based Case-Control Study"

_ijerph, 2019, doi:10.3390/ijerph16040617_

Round 1

Reviewer 1 Report

Comments to the authors

This study describes the risk factors associated with dengue virus infection in Guangdong Province where the cases and controls were identified from sero-epidemiologic samples from community. The study aimed to identify the environmental and socio-epidemiologic risk factors associated with either symptomatic or asymptomatic dengue infection. Overall, the study was well-aimed, however, there are several major points that need to be addressed, and also require to restructure the manuscript.

Major points:

1.      My main concern is selection of cases and controls. Authors identified cases and controls based on positive IgM and IgG from single serum samples which actually is regarded as ‘probable dengue infection’ and introduced misclassification of cases. This single serum sample should be coupled with a minimum of one second confirmatory test to confirm the dengue infection. As such, this study described the risk factors associated with ‘probable dengue infection’. It is highly encouraged to perform the second confirmatory test on this selected study population to confirm the dengue infection.

2.      Authors described controls were selected by frequency matching on age and sex. However, no description was made on the matching age categories (or age as continuous variable) and which percentage is matched to what categories.

3.      L115 – what does it mean by “further match”? How did authors select 237 controls from 308 individuals who completed questionnaires?

4.      L118 – how did authors select the subjects for either face-to-face interview or phone call? Due to differences in mode of administering questionnaires, it is introducing information bias. How did authors deal with this?

5.      L119 - please explain clearly the definitions of questions that were asked in methods section – such as how authors defined activities in the park, outdoor sports, contacting with dengue fever patients, etc.

6.      L128 onwards - authors need to explain in details for statistical analysis in methods section.

a.       How did authors derive multivariable models? Which variables are selected to include and why? What is the method employed for variables selection? How did authors choose best fit model?

b.      It is important to add not only the variables that are statistically significant but also the variables that have clinical/ social importance such as age, gender, etc. in the multivariable models.

c.       What is the level of statistical significance?

d.      Did authors need to use Fisher’s exact test or equivalent in place of chi-squared test? If so, please describe.

e.       “Unconditional” logistic regression is a better term. Authors still need to control for the matching variables in multivariable logistic regression models due to frequency matching design.

7.      Abstract L11 and introduction L51 – it is not correct to say that there is no vaccine to prevent dengue fever so far. It may be true that the vaccine is not approved in China yet. There is Dengvaxia vaccination. Please correct these statements and reference accordingly.

8.      Please draw a flow diagram to explain the participant selection from source population (refer to STARD 2015 flow diagram) to aid in results section.

9.      Table 1 – If authors frequency-matched the age as categorical variable, please describe as category.

10.  Table 2 – there are several variables that need to be explained. For example, contacting with dengue patients. Was it defined as current contact or past contact? If it is past how many days/ weeks ago? Mosquito repellents – was it any repellents or spray or patch? How did authors define occasional and often (what is the frequency cut-off)? Authors can consider describing full details in methods instead of table 2 footnote.

11.  Table 3 – B coefficient and constant are not required to present when there is odds ratio which is more relevant for this study. Please report P value 0.000 as P<0.001.

12.  L189-191 – the way the residents dress in the park is a common thinking and it would be affected by seasonal variation. Authors should discuss this socio-behavioural impact to your study outcome (dengue infection) using a behavioural model or observational findings from previous studies.

13.  L232-235 – this another explanation is incorrect.

14.  Conclusion needs to be rewritten to a great degree.

Minor points:

1.      It is highly recommended to use STROBE checklist of case-control study to report for proper reporting structure. Please reorganize the structure of manuscript accordingly. https://www.strobe-statement.org/fileadmin/Strobe/uploads/checklists/STROBE_checklist_v4_case-control.pdf

2.      L81 - Brief description of the demographic information of the database in this present manuscript will be helpful to readers.

3.      L81 – Authors described 3,136 samples although the total from 4 regions add up to 3,316. Please recheck. Was the selection from 4 regions proportional to population density in each region?

4.      L82 - Please express the number as following example: Yuexiu District in Guangzhou city (n=699).

5.      L107 to 109 – statistical difference paragraph should be in results section, not in the methods.

6.      L109 - Please express the P value simply as (P=0.950) without >0.05, the same goes to the whole manuscript.

7.      L137 – Replace with demographic characteristics in the place of potential confounders.

8.      L166-169 – these sentences belong to methods.

9.      Multivariable analysis - It is critical to describe not just the odds ratio, also 95% CI for all ORs.

10.  L174-178 – The equation is quite unnecessary considering the study’s aim.

11.  Introduction can be improved for better description. Please clearly state the aims and objectives of the study at the last paragraph of introduction.

12.  L66-71 – Explanation of geographical position of the study location can be placed before study aim and objective and briefed.

13.  L183 and throughout discussion – Authors performed logistic regression which provided Odds Ratio. Thus, in the sentence “increase 1.70-time risk of developing dengue infection” is inaccurate. Authors should replace odds/probability in place of risk since we are talking about odds here, not risk (which are different effect estimates).

14.  L207 – one room with three persons and above were shared by the parents and their young children – Do authors have data to support this claim?

15.  L262 – to reduce the bias, some of the questionnaires were completed by the doctors ….. I am unclear with how does it reduce the bias. Please explain more.

16.  Please discuss the generalizability of this study to other communities.

17.  Authors discussed Aedes albopictus throughout the manuscript but not Aedes aegypti which is the major vector. Is it because A. albopictus population dominate in the study location? If so, please discuss provide relevant references.

18.  Please consider attaching questionnaire in additional file for readers to refer.

19.  Univariable and multivariable are preferred terms in place of univariate and multivariate.

20.  The correct term is “odds ratio”, not “odd ratio”.

21.  L109 - Please express the P value simply as (P=0.950) without >0.05

22.  English can be improved throughout.

Author Response

Major points:

1. My main concern is selection of cases and controls. Authors identified cases and controls based on positive IgM and IgG from single serum samples which actually is regarded as ‘probable dengue infection’ and introduced misclassification of cases. This single serum sample should be coupled with a minimum of one second confirmatory test to confirm the dengue infection. As such, this study described the risk factors associated with ‘probable dengue infection’. It is highly encouraged to perform the second confirmatory test on this selected study population to confirm the dengue infection.

Response: Thank you so much for your pretty good comments. In our study, we used the enzyme-linked immunosorbent assay (ELISA) to detect dengue antibody IgG and IgM. Currently, ELISA has been widely considered the most commonly used test for dengue infection diagnosis due to its high sensitivity and feasibility. And the seroconversion of IgM or IgG indicates dengue infection. In addition, the undefined results were confirmed by the colloidal gold method. However, the nucleic acid testing method can only detect the status of current infection in the residents, not the status of previous infection. Therefore, we think that the individuals detected as IgG antibody positive and IgM antibody positive in this study can be considered as the cases of dengue infection.

2. Authors described controls were selected by frequency matching on age and sex. However, no description was made on the matching age categories (or age as continuous variable) and which percentage is matched to what categories.

Response: Thank you so much for this comment. Indeed, the age should be described as categorical variable. We added the corresponding content in Table 1 in the revised version.

3. L115 – what does it mean by “further match”? How did authors select 237 controls from 308 individuals who completed questionnaires?

Response: Thanks for the comment. Just as descripted in the manuscript, according to the community information of the 237 cases, a further match was done and 237 controls were selected. For example, when the age of one female case is 45 years old, the age of two healthy women both are 45 years old and can be considered as the case’s control. In order to choose a more comparable control, we choose the woman who lived in the same community as the case. It was modified appropriately to present a simple understanding for the readers in line 582.

4. L118 – how did authors select the subjects for either face-to-face interview or phone call? Due to differences in mode of administering questionnaires, it is introducing information bias. How did authors deal with this?

Response: Thanks for your advice. The phone questionnaire was carried out by these trained investigators to obtain the information of the cases and controls. And the subjects who interrupted the telephone and whose information filled in the questionnaire had some logical errors were interviewed face to face to verify the integrity and validity of their information (line 588). Therefore, the same treatment was used in both the case group and the control group, so the information bias did not need to be considered.

5. L119 - please explain clearly the definitions of questions that were asked in methods section – such as how authors defined activities in the park, outdoor sports, contacting with dengue fever patients, etc.

Response: Thanks for your suggestion very much. We explained the definitions of activities in the park, outdoor sports according to subtropical monsoon climate in Guangdong province the leisure habits of the natives. And we have added the definition of contacting with dengue fever patients. And we have moved the definitions from the footnote of the table 2 to the method section (line 600-609).

6. L128 onwards - authors need to explain in details for statistical analysis in methods section.

a. How did authors derive multivariable models? Which variables are selected to include and why? What is the method employed for variables selection? How did authors choose best fit model?

Response: Thank you so much for this comment. We have added the related details in the methods section (line 612-678).

b. It is important to add not only the variables that are statistically significant but also the variables that have clinical/ social importance such as age, gender, etc. in the multivariable models.

Response: Thanks for your suggestion. We tried to add the variables of age and gender to the multivariable model, but the results did not change. Besides, the variables of age and gender were conducted as the matching variables in our study, so they didn’t need to be imported in the multivariable models. In order to avoid missing important factors, p<0.1 was set as the significance level of the univariate analysis. Then the housing location variable and the living floor variable were also added to the multivariable models.

c. What is the level of statistical significance?

Response: Thank you for the good comment. We have added the word “p < 0.05 was set as the significance level of the c2 test and the multivariate analysis. However, in order to avoid missing important factors, p<0.1 was set as the significance level of the univariate analysis.” to the manuscript (line 678-681).

d. Did authors need to use Fisher’s exact test or equivalent in place of chi-squared test? If so, please describe.

Response: Thank you for the suggestion. In this study, we didn’t need to use Fisher’s exact test or equivalent in place of chi-squared test.

e. “Unconditional” logistic regression is a better term. Authors still need to control for the matching variables in multivariable logistic regression models due to frequency matching design.

Response: Thank you so much for this comment. We tried to add the variables of age and gender to the multivariable model, but the results did not change. And the matching variables made the case group and the control group equally comparable, so we thought that they didn’t need to enter the multivariable logistic regression models.

7. Abstract L11 and introduction L51 – it is not correct to say that there is no vaccine to prevent dengue fever so far. It may be true that the vaccine is not approved in China yet. There is Dengvaxia vaccination. Please correct these statements and reference accordingly.

Response: Thank you for the suggestion. We have changed the words as the word “the absence of a licensed vaccination in China” according to your comments (line 12 and line 204).

8. Please draw a flow diagram to explain the participant selection from source population (refer to STARD 2015 flow diagram) to aid in results section.

Response: That’s very nice of you, we have added a flow diagram (Fig. 1) to explain the participant selection (line 695).

9. Table 1 – If authors frequency-matched the age as categorical variable, please describe as category.

Response: Thank you so much for this comment. We have described the age as categorical variable in table 1.

10. Table 2 – there are several variables that need to be explained. For example, contacting with dengue patients. Was it defined as current contact or past contact? If it is past how many days/ weeks ago? Mosquito repellents – was it any repellents or spray or patch? How did authors define occasional and often (what is the frequency cut-off)? Authors can consider describing full details in methods instead of table 2 footnote.

Response: Thanks for this comment. we have added the definition of contacting with dengue patients, mosquito repellents, occasionally and often (line 600-609).

11. Table 3 – B coefficient and constant are not required to present when there is odds ratio which is more relevant for this study. Please report P value 0.000 as P<0.001.

Response: Thanks for your comment. B coefficient and constant were removed from table 3. Besides we have changed P value 0.000 as P<0.001 in table 3.

12. L189-191 – the way the residents dress in the park is a common thinking and it would be affected by seasonal variation. Authors should discuss this socio-behavioural impact to your study outcome (dengue infection) using a behavioural model or observational findings from previous studies.

Response: Thanks for your suggestion. Guangdong's climate belongs to the typical monsoon marine climate in the south subtropical zone, which is characterized by abundant light and heat, small temperature difference and long summer. Accordingly, the residents who like to have activities in the park prefer dressing short sleeved shirts and shorts. In addition, p<0.1 was set as the significance level of the univariate analysis according to your suggestion. Then the housing location variable and the living floor variable were also added to the multivariable models. As a result, the Variable of activities in the park was not in the unconditional multivariate Logistic regression equation, the interpretation about activities in the park is deleted in the discussion part.

13. L232-235 – this another explanation is incorrect.

Response: Thanks for the comment. After consulting the relevant literature and other information, we found a reasonable explanation and changed the word to the word “Other studies found that mosquito nets play a protective role in preventing dengue virus infection in rural settings [29]. However, we recruited our study population from urban areas.” (line 1247-1249).

14. Conclusion needs to be rewritten to a great degree.

Response: That’s very nice of you, we have rewritten the conclusion according to your comments (line 1523-1530).

Minor points:

1. It is highly recommended to use STROBE checklist of case-control study to report for proper reporting structure. Please reorganize the structure of manuscript accordingly. https://www.strobe-statement.org/fileadmin/Strobe/uploads/checklists/STROBE_checklist_v4_case-control.pdf

Response: Thanks for the comment. We have reorganized the structure of manuscript according to the STROBE_checklist_v4_case-control.pdf.

2. L81 - Brief description of the demographic information of the database in this present manuscript will be helpful to readers.

Response: Thanks for your comment. We have made a brief description of the demographic information of the database, and have moved the related information from the method section to the result section (line 687-689).

3. L81 – Authors described 3,136 samples although the total from 4 regions add up to 3,316. Please recheck. Was the selection from 4 regions proportional to population density in each region?

Response: Thanks for the comment. We have checked the data again and corrected the mistakes (line 685-686).

4. L82 - Please express the number as following example: Yuexiu District in Guangzhou city (n=699).

Response: That’s very nice of you, we have changed the expression according to your comments.

5. L107 to 109 – statistical difference paragraph should be in results section, not in the methods.

Response: Thanks for the comment. We have moved the statistical difference paragraph from the method section to the result section (line 691-693).

6. L109 - Please express the P value simply as (P=0.950) without >0.05, the same goes to the whole manuscript.

Response: Thanks for the comment. We have expressed the P value simply without >0.05 or <0.1 in the whole manuscript.

7. L137 – Replace with demographic characteristics in the place of potential confounders.

Response: Thank you for the comment, we have changed the expression according to your comments.

8. L166-169 – these sentences belong to methods.

Response: Thanks for the comment. We have moved these sentences from the result section to the method section.

9. Multivariable analysis - It is critical to describe not just the odds ratio, also 95% CI for all ORs.

Response: Thank you for the comment, we have added 95% CI according to your comments.

10. L174-178 – The equation is quite unnecessary considering the study’s aim.

Response: Thanks for the comments. The equation is deleted.

11.Introduction can be improved for better description. Please clearly state the aims and objectives of the study at the last paragraph of introduction.

Response: Thanks for the comments. We reorganized the structure of the introduction section according to your suggestion.

12. L66-71 – Explanation of geographical position of the study location can be placed before study aim and objective and briefed.

Response: Thanks for the comment. But, after considering another reviewer's suggestions we decided to place the explanation of geographical position of the study location in the materials and methods section (line 222-226).

13.L183 and throughout discussion – Authors performed logistic regression which provided Odds Ratio. Thus, in the sentence “increase 1.70-time risk of developing dengue infection” is inaccurate. Authors should replace odds/probability in place of risk since we are talking about odds here, not risk (which are different effect estimates).

Response: Thanks for the comments. We have replaced probability in place of risk (line 920).

14. L207 – one room with three persons and above were shared by the parents and their young children – Do authors have data to support this claim?

Response: Thanks for the comments. It was just one of possible explanations of the result that more than one person per room was a protective factor. And our study was conducted in the communities, where the individuals lived together in family groups, so one room with three persons and above were commonly shared by the parents and their young children. In addition, other study (Wang HS , Huang XN , Jiang JX, et al. Sleep location in Chinese children aged 05 years old[J]. Chinese Journal of Child Health Care, 2008, 16, 420-422.) showed that 62.48% of Chinese children aged 0-5 years old took bed sharing with their parents. Hence this claim can be supported (line 1222-1227).

15. L262 – to reduce the bias, some of the questionnaires were completed by the doctors ….. I am unclear with how does it reduce the bias. Please explain more.

Response: Thanks for the valuable advice. After discussion with other co-authors, we thought the measures that some of the questionnaires were completed by the community doctors, and some of the questionnaires were completed by face-to-face interviews could not reduce the bias. At last, the sentences were deleted.

16. Please discuss the generalizability of this study to other communities.

Response: Thanks for the comment. The generalizability of this study has been added in the limitation section in the revised version (line 1514-1521).

17. Authors discussed Aedes albopictus throughout the manuscript but not Aedes aegypti which is the major vector. Is it because A. albopictus population dominate in the study location? If so, please discuss provide relevant references.

Response: Thank you for the comment, Aedes albopictus, also known as the Asian tiger mosquito, is an indigenous species and the predominant vector of dengue fever in China. And Aedes albopictus was the dominant species in Guangdong Province, which widely distributed and had high density (line 102-103). We have added the relevant references to the manuscript. Such as:

20.   Guo Y , Song Z , Luo L , et al. Molecular evidence for new sympatric cryptic species of Aedes albopictus (Diptera: Culicidae) in China: A new threat from Aedes albopictus subgroup? Parasites & Vectors, 2018, 11, 228; Doi: 10.1186/s13071-018-2814-8.

21.   De-Sheng J , Wei-Long T , Chang-Jun W , et al. Aedes-borne diseases prevention and control. Chinese Journal of Hygienic Insecticides & Equipments, 2017, 23, 1-7.

18. Please consider attaching questionnaire in additional file for readers to refer.

Response: Thanks for your comments. We are agreeable to attach questionnaire in additional file for readers to refer as part of this publication.

19. Univariable and multivariable are preferred terms in place of univariate and multivariate.

Response: Thank you for the comment, we found the same expression in many relevant references, such as: Snydermackler N , Joaquín Sanz, Kohn J N , et al. Social status alters immune regulation and response to infection in macaques. Science, 2016, 291:1041-1045. Doi: 10.1126/science.aah3580.

So we thought “univariate” and “multivariate” are also the correct expressions.

20. The correct term is “odds ratio”, not “odd ratio”.

Response: Thanks for the comment. We have changed “odd ratio” to “odds ratio”.  

21. L109 - Please express the P value simply as (P=0.950) without >0.05

Response: Thanks for the comment. We have expressed the P value simply without >0.05 or <0.1 in the whole manuscript.

22. English can be improved throughout.

Response: Thanks for the comment. We have revised the whole manuscript carefully and tried to avoid any grammar or syntax error. In addition, we have asked a native speaker to check the English. We believe that the language is now acceptable for the review process.

Reviewer 2 Report

Liu and colleagues present us with a case-control study to explore the individual risk factors for dengue virus infection in a Chinese province, as well as provide a scientific basis for the prevention and supervision of dengue in the future.

In general, the study has a number of issues to improve and clarify. The design of a case-control study in this case is unclear. The goal is too broad and has not been reached in its entirety, there are errors of argumentation and minor corrections are required in English.

I then tried to summarize more focussed comments that made me have this opinion about the study.

Introduction

-You have a lot of information. It is disorganized, which makes reading tiring and uninteresting. It needs a general reorganization. For example, the authors begin by citing information on the behavior of dengue in Asia, then in some Asian countries (China, Sri Lanka ...), after Guangdong province. Soon after, they are talking about other data in China and other provinces. Therefore, there is no linearity of thought;

-The introduction has some conflicting information, especially in the third paragraph. I believe that after a better reorganization the authors will be able to overcome this;

-The point is not clear. Nor the hypotheses. Please clarify and highlight the same

- The third paragraph is giant, this is not well seen from the point of view of style;

-The third paragraph should be completely reformulated. Characterization of the study site should be part of the materials and methods section and not the introduction;

Method

- It is not clear how a database of a work on ISTs was used for this research;

-Using a base with IgG positivity information is not the most appropriate and this can seriously compromise the screening and search results

-It would be interesting a flow chart of case selection;

-Did some subjects have the information checked in person? Which are? According to which criterion?

-The authors present association data in the method section and this is not correct in my opinion. Moreover, I believe that this information is of little use in this context.

Results

-I believe that the categorizations used in table 02 influenced the p-value. I do not know what the authors used to classify each variable, but I imagine the authors know the weight of this classification.

-In addition, putting reference category in the bivariate analysis to me is unusual. I would like to hear the authors about this.

-I ask you to clarify which tests you used to achieve p-value, especially for those variables where there were values below 05 in each casela / sub-category.

-Has the definitions of Actives in the Park, Having outdoor sports and Good Ventilation effect been based on the literature?

Discussion

-The discussion is interesting since the authors wisely seek explanations for the results of the multivariate analysis. However, it is not enough, because it loses its capacity and comparison among other studies in a different reality. In this sense, it is necessary to reformulate it; In the way it is placed, it serves only to reinforce what is already known.

Author Response

Introduction

-You have a lot of information. It is disorganized, which makes reading tiring and uninteresting. It needs a general reorganization. For example, the authors begin by citing information on the behavior of dengue in Asia, then in some Asian countries (China, Sri Lanka ...), after Guangdong province. Soon after, they are talking about other data in China and other provinces. Therefore, there is no linearity of thought;

-The introduction has some conflicting information, especially in the third paragraph. I believe that after a better reorganization the authors will be able to overcome this;

-The point is not clear. Nor the hypotheses. Please clarify and highlight the same

- The third paragraph is giant, this is not well seen from the point of view of style;

-The third paragraph should be completely reformulated. Characterization of the study site should be part of the materials and methods section and not the introduction;

Response: That’s very nice of you. We reorganized the structure of the introduction section according to your suggestion (line 31-219).

Method

- It is not clear how a database of a work on ISTs was used for this research;

Response: Thank you for your comment. The description of 200,000-sample database was given to let the readers had a simple understanding of the data source of the study. Among the 200,000 samples, the male to female ratio was 0.47:1. The proportions of age groups were 16.68%, 18.1%, 36.63% and 28.59% (under 19 years old, 19 to 40 years old, 41 to 65 years old, and above 65 years old, respectively). In addition, 7.6% of the participants were illiterate, and the percentages of the individuals with other education degrees (primary, junior high school, senior high school and diploma and over) were 32.27%, 18.89%, 20.04% and 6.71%, respectively.

-Using a base with IgG positivity information is not the most appropriate and this can seriously compromise the screening and search results

Response: Thank you so much for your pretty good comments. In our study, we used the enzyme-linked immunosorbent assay (ELISA) to detect dengue antibody IgG and IgM. Currently, ELISA has been widely considered the most commonly used test for dengue infection diagnosis due to its high sensitivity and feasibility. And the seroconversion of IgM or IgG indicates dengue infection. In addition, the undefined results were confirmed by the colloidal gold method. Therefore, we think that the individuals detected as IgG antibody positive and IgM antibody positive in this study can be considered as the cases of dengue infection.

-It would be interesting a flow chart of case selection;

Response: Thank you for your comment, we have added a flow diagram (Fig. 1) to explain the participant selection (line 695).

-Did some subjects have the information checked in person? Which are? According to which criterion?

Response: This is a good comment. In this study, the subjects who interrupted the telephone and whose information filled in the questionnaire had some logical errors were interviewed face to face to verify the integrity and validity of their information (line 585).

-The authors present association data in the method section and this is not correct in my opinion. Moreover, I believe that this information is of little use in this context.

Response: Thanks for your comment. In our study, there was no statistical difference between the persons who were willing to receive questionnaire survey and those who were unwilling in gender and age. Then the results suggested that the participants had certain representation to the residents in Guangdong province. And the related content has moved the related information from the method section to the result section (line 691-693).

Results

-I believe that the categorizations used in table 02 influenced the p-value. I do not know what the authors used to classify each variable, but I imagine the authors know the weight of this classification.

Response: Thanks for your comment. We classified each variable according to the references and the actual situation of Guangdong.

-In addition, putting reference category in the bivariate analysis to me is unusual. I would like to hear the authors about this.

Response: Thanks for your comment. Because in this study, the investigated variables were analyzed by unvariate and multivariate un-conditioned Logistic regression model, so the reference category could be got. And the same or similar expression could be found in some references, such as: 23 Velascosalas ZI, Sierra GM, Guzmán DM et al. Dengue Seroprevalence and Risk Factors for Past and Recent Viral Transmission in Venezuela: A Comprehensive Community-Based Study. American Journal of Tropical Medicine & Hygiene. 2014, 91, 1039-1048; Doi: 10.4269/ajtmh.14-0127.

-I ask you to clarify which tests you used to achieve p-value, especially for those variables where there were values below 05 in each casela / sub-category.

Response: Thanks for your comment. In order to test for differences in demographic characteristics between cases and controls, c2 test was used. A univariate unconditioned logistic regression analysis was applied for preliminary screening of these variables in the questionnaire, except the variables in general demographic characteristics.

-Has the definitions of Actives in the Park, Having outdoor sports and Good Ventilation effect been based on the literature?

Response: Thanks for your comment. It’s a pity that the literatures related to the definitions are not found. But the definitions of Actives in the Park, Having outdoor sports and Good Ventilation effect are defined according to local conditions in Guangdong province and the relevant information which were consulted carefully (line 600-609).

Discussion

-The discussion is interesting since the authors wisely seek explanations for the results of the multivariate analysis. However, it is not enough, because it loses its capacity and comparison among other studies in a different reality. In this sense, it is necessary to reformulate it; In the way it is placed, it serves only to reinforce what is already known.

Response: Thanks for your comment. Several sentences have been reformulated in the discussion section in the revised version.

Round 2

Reviewer 1 Report

Thank you for revising the manuscript which has been vastly improved. However, further improvements are required.

1.      Abstract – Please provide how authors define cases and controls for the study.

2.      L104 – Please describe/ spell out the age categories authors use to frequency-match.

3.      L106 – Authors described that participants volunteered to become part of the samples. As such, this introduces a self-selection bias or otherwise known as volunteer bias. The information provided by the volunteered participants might be different from the general population. Please discuss this in limitations.

4.      L123 – Thank you for including this paragraph which makes reader to understand the study better. But please describe the time factor when authors defined “contact with DF patients” – for example, past living or working experience with DF patients – was it defined as in the past 1 month/ 6 months or no time frame was mentioned?

5.      L154-156 – In this sentence there are two p values which I suppose one for gender and one for age. If it is so, please place the corresponding p value immediately next to gender and age. It became unclear when the two p values are placed next to age and end of the sentence.

6.      Table 1 – Age categories – Is it ≥15 years or ≤15 years?

7.      Multivariable analysis – as a rule of frequency matching design, you still need to control for matching variables in unconditional logistic regression analysis. This is because frequency matching makes the controls more similar in distribution compared to cases in terms of matching variables. As a result, odds ratio will be biased towards the null (i.e., OR = 1). Please refer to Rothman KJ, Modern Epidemiology. Thus, please include matching variables in multivariable analysis although they are not statistically different. Please note that there is difference in terms of statistical analysis method between frequency matching and individual matching design.

8.      L172 – Authors stated that there is no statistical significant differences in housing type (P=0.040), ……, average no. of person per room (P < 0.001), ….. However, P values suggest significant difference. Please thoroughly rewrite.

9.      L175 – Please include P value for each variable although they are not significant, for example garbage collection sites (P = 0.681) and so on.

10.  L143 – odds ratio, not odds rations

Author Response

Comments and Suggestions for Authors

Thank you for revising the manuscript which has been vastly improved. However, further improvements are required.

1. Abstract – Please provide how authors define cases and controls for the study.

Response: Thank you so much for your pretty good comments. We have added the definitions of cases and controls in the abstract section (line 14-19).

2. L104 – Please describe/ spell out the age categories authors use to frequency-match.

Response: Thank you so much for this comment. We have added the related details in the methods section (line 112-114).

3. L106 – Authors described that participants volunteered to become part of the samples. As such, this introduces a self-selection bias or otherwise known as volunteer bias. The information provided by the volunteered participants might be different from the general population. Please discuss this in limitations.

Response: Thanks for your suggestion. In this study, we found that no statistical difference in gender (p = 0.950) and age (p = 0.127) existed between persons who were willing to receive the questionnaire survey and those who were unwilling to receive the questionnaire survey, then we thought that the volunteered participants had a good representation. However, volunteer bias was difficult to avoid completely, so we discuss the volunteer bias in limitations according to your suggestion (line 288-290).

4. L123 – Thank you for including this paragraph which makes reader to understand the study better. But please describe the time factor when authors defined “contact with DF patients” – for example, past living or working experience with DF patients – was it defined as in the past 1 month/ 6 months or no time frame was mentioned?

Response: Thank you so much for this comment. “Contact with patients with DF” was defined as previous living or working experience with patients with DF in the past years in his/her life (line 131).

5. L154-156 – In this sentence there are two p values which I suppose one for gender and one for age. If it is so, please place the corresponding p value immediately next to gender and age. It became unclear when the two p values are placed next to age and end of the sentence.

Response: Thank you for the comment, we have changed the sentence as the sentence “No statistical difference in gender (p = 0.950) and age (p = 0.127) existed between persons who were willing to receive the questionnaire survey and those who were unwilling to receive the questionnaire survey” (line 166-168).

6. Table 1 – Age categories – Is it ≥15 years or ≤15 years?

Response: Thanks for the comments. The age group should be ≤15 years. We corrected this mistake in Table 1.

7. Multivariable analysis – as a rule of frequency matching design, you still need to control for matching variables in unconditional logistic regression analysis. This is because frequency matching makes the controls more similar in distribution compared to cases in terms of matching variables. As a result, odds ratio will be biased towards the null (i.e., OR = 1). Please refer to Rothman KJ, Modern Epidemiology. Thus, please include matching variables in multivariable analysis although they are not statistically different. Please note that there is difference in terms of statistical analysis method between frequency matching and individual matching design.

Response: Thank you so much for this comment. We controlled for the age, gender and community information variables in multivariable logistic regression models due to frequency matching design, then the results of multivariate analysis were the same as before (line 151-152, line 205).

8. L172 – Authors stated that there is no statistical significant differences in housing type (P=0.040), ……, average no. of person per room (P < 0.001), ….. However, P values suggest significant difference. Please thoroughly rewrite.

Response: Thanks for the comment. We have rewritten this sentence and corrected the mistakes (line 188-191).

9. L175 – Please include P value for each variable although they are not significant, for example garbage collection sites (P = 0.681) and so on.

Response: Thank you for the comment, we have added the P value n according to your comments (line 197-199).

10. L143 – odds ratio, not odds rations

Response: Thanks for the comment. We have changed “odds rations” to “odds ratios” (line 155).

Reviewer 2 Report

Thanks for considering the suggestions. In fact, the current version is much more robust and organized. However, I would still like to point out some issues.

Method

1. Please clarify the information about pairing for "age and sex ratio" according to the categorization used;

2. Very interesting that the authors have made a section only of definitions. This in fact helps to better understand the research and propitiates the replication of the study in other realities, as well as a basis for comparison among other studies. This was not the case in the previous version. I would ask the authors, however, to clarify some points:

a-I am not convinced that "domestic sewage and garbage management and participation in community hygiene management interventions" can be considered personal hygiene, which in fact refers to something more personal in the sense of caring for oneself;

b - "Contact with patients with DF". Do the authors understand contact as a living? Was the various "time" considered?

c - Some definitions are extremely technical like "an open-ventilation area of not less than 5% of the floor area of each domicile". Are the authors sure that the information collected was understood by the respondent? Or would that be a limitation?Linha 145- Erro de grafia: odds ratio e não ration

Results

1. There are two crosses (sex and age) and only one p value (0.127). Was it the same value for both?

2. R-172 to 175. Some variables presented p value <0.05 and yet the authors cite as non-significant. It makes no sense to me;

3.Table 01. Is it possible to tell in footnote the value of the current minimum wage? I believe that it is interesting to have a notion of the income of the population, since this is an important variable;

Discussion

1. Line 206- "Our conclusion contradicts that of Velascosalas et al. [23] ". Would not their discoveries be more appropriate? Another detail. Why does it contradict? What is the conclusion of Velascosalas? Does not make sense.

Author Response

Comments and Suggestions for Authors

Thanks for considering the suggestions. In fact, the current version is much more robust and organized. However, I would still like to point out some issues.

Method

1. Please clarify the information about pairing for "age and sex ratio" according to the categorization used;

Response: Thank you so much for this comment. We have added the related details in the methods section (line 109-111).

2. Very interesting that the authors have made a section only of definitions. This in fact helps to better understand the research and propitiates the replication of the study in other realities, as well as a basis for comparison among other studies. This was not the case in the previous version. I would ask the authors, however, to clarify some points:

a-I am not convinced that "domestic sewage and garbage management and participation in community hygiene management interventions" can be considered personal hygiene, which in fact refers to something more personal in the sense of caring for oneself;

Response: Thanks for the comment. We thought that "domestic sewage and garbage management and participation in community hygiene management interventions" can be considered sanitary behavior. We changed “personal hygiene habits” to “environmental sanitation” (line 69, line 123, line 183).

b - "Contact with patients with DF". Do the authors understand contact as a living? Was the various "time" considered?

Response: Thank you so much for this comment. “Contact with patients with DF” was defined as previous living or working experience with patients with DF in the past years in his/her life (line 131).

c - Some definitions are extremely technical like "an open-ventilation area of not less than 5% of the floor area of each domicile". Are the authors sure that the information collected was understood by the respondent? Or would that be a limitation?Linha 145- Erro de grafia: odds ratio e não ration

Response: Thanks for the comment. The definition of “good ventilation” was given in our questionnaire, however, the information was obtained according to the respondents’ subjective feelings. Besides, we discuss the limitation in limitations according to your suggestion (line 290-293). And We have changed “odds rations” to “odds ratios” (line 155).

Results

1. There are two crosses (sex and age) and only one p value (0.127). Was it the same value for both?

Response: Thank you for the comment, we have changed the sentence as the sentence “No statistical difference in gender (p = 0.950) and age (p = 0.127) existed between persons who were willing to receive the questionnaire survey and those who were unwilling to receive the questionnaire survey” (line 166-168).

2. R-172 to 175. Some variables presented p value <0.05 and yet the authors cite as non-significant. It makes no sense to me;

Response: Thanks for the comment. We have rewritten this sentence and corrected the mistakes (line 188-191).

3.Table 01. Is it possible to tell in footnote the value of the current minimum wage? I believe that it is interesting to have a notion of the income of the population, since this is an important variable;

Response: Thanks for the comment. We added the related contents “The minimum wage in Guangzhou City is 2,100 yuan per month. And the minimum wage in Zhongshan City is 1,720 yuan per month” in footnote of table 1.

Discussion

1. Line 206- "Our conclusion contradicts that of Velascosalas et al. [23] ". Would not their discoveries be more appropriate? Another detail. Why does it contradict? What is the conclusion of Velascosalas? Does not make sense.

Response: Thank you for the comment. We used the sentence “On the contrary, Velascosalas et al found that crowded household with more than 1.5 occupants in one room were risk factor of dengue infection” instead of “Our conclusion contradicts that of Velascosalas et al.” according to your comments (line 230-231).